# Efficacy, Tolerability, and Safety of Toludesvenlafaxine for the Treatment of Major Depressive Disorder—A Narrative Review

**DOI:** 10.3390/ph16030411

**Published:** 2023-03-08

**Authors:** Octavian Vasiliu

**Affiliations:** Department of Psychiatry, Dr. Carol Davila University Emergency Central Military Hospital, 010825 Bucharest, Romania; octavvasiliu@yahoo.com

**Keywords:** toludesvenlafaxine, ansofaxine, LY03005, LPM570065, triple monoamine reuptake inhibitors, major depressive disorders, efficacy, tolerability, treatment-resistant depression, first-in-class antidepressant

## Abstract

The estimated rate of treatment-resistant major depressive disorder (TRD) remains higher than 30%, even after the discovery of multiple classes of antidepressants in the last 7 decades. Toludesvenlafaxine (ansofaxine, LY03005, or LPM570065) is a first-in-class triple monoaminergic reuptake inhibitor (TRI) that has reached clinical use. The objective of this narrative review was to summarize clinical and preclinical evidence about the efficacy, tolerability, and safety of toludesvenlafaxine. Based on the results of 17 reports retrieved in the literature, the safety and tolerability profiles of toludesvenlafaxine were good in all clinical trials, and the pharmacokinetic parameters were well described in the phase 1 trials. The efficacy of toludesvenlafaxine was demonstrated in one phase 2 and one phase 3 trial, both on primary and secondary outcomes. In conclusion, this review highlights the favorable clinical results of toludesvenlafaxine in only two short-term trials that enrolled patients with major depressive disorder (MDD) (efficacy and tolerability were good for up to eight weeks), indicating the need for more good quality, larger-sample, and longer-term trials. Exploring new antidepressants, such as TRI, can be considered a priority for clinical research due to the high rates of TRD, but also due to the significant percentages of relapse in patients with MDD.

## 1. Introduction

In the context of the constant debate in the literature about the unmet needs of patients with major depressive disorder (MDD), which usually revolves around the high incidence of treatment-resistant depression (TRD) and the negative functional consequences of this resistance [1], the urgency of finding new antidepressants becomes obvious. Another important aspect in the treatment of MDD and especially TRD is represented by the “antidepressant polypharmacy” (i.e., concomitant administration of ≥2 antidepressants), a strategy considered to improve the outcomes by activating multiple neurotransmitter pathways at the same time [2]. Combining multiple antidepressants or adding different agents to the ongoing antidepressant in order to improve its efficacy or to speed its onset is not without risks, and adverse events might appear more often in these patients, especially in vulnerable populations (i.e., patients with late-life depression, multiple somatic or psychiatric comorbidities, etc.) [3,4,5,6,7]. The risk of negative pharmacokinetic and pharmacodynamic interactions also needs to be considered when multiple medications are concomitantly administered in patients with MDD, potentially triggering more frequent treatment tolerability and tolerance problems than in the case of monotherapy [5,8,9,10].

The monoamine hypothesis was launched in the 1950s, and it stipulates that the deficit of serotonin, dopamine, and norepinephrine in the central nervous system (CNS) is the pathophysiological basis of depression [11,12,13]. Based on this hypothesis, antidepressants can raise the levels of monoamine neurotransmitters in the brain and, as a consequence, effectively decrease the severity of depressive symptoms [11]. *Serotonin* has been presumed to have a causal role in the pathogenesis of depression based on multiple pieces of evidence: tryptophan depletion, which is an acute dietary manipulation, leads to lower CNS serotonin activity and may produce brief, but clinically significant, depressive symptoms in recovered depressed patients; peripheral inflammation has been involved in the onset of depression in vulnerable individuals by lowering plasma tryptophan level; serotoninergic antidepressants have a long history of clinical use, and they have traditionally been associated with serotonin pathway activation; and serotonin deficiency has been related to anxiety, obsessions, and compulsions [11,12,13,14,15,16,17,18,19]. *Norepinephrine* is considered a key element in the pathogenesis of MDD because locus coeruleus is the major anatomical target for monoaminergic antidepressants; a rapid decrease in the catecholamine level in the CNS has been associated with relapses of MDD; the mechanisms of the inhibition of monoamine oxidase, antagonism of presynaptic norepinephrine receptors, and inhibition of norepinephrine reuptake from the synaptic cleft increase the catecholamine neurotransmission and improve the depressive symptoms; and norepinephrine deficits have been related to the onset of low energy, inattention, executive dysfunction, and decreased alertness [20,21,22,23]. *Dopamine* neurotransmission dysfunctions are considered responsible for core symptoms of depression, e.g., anhedonia and cognitive deficits; translational and clinical studies demonstrate deficits of the dopaminergic system in depression; and antidepressants enhancing dopaminergic transmission improved symptoms of energy, pleasure, and interest in MDD patients [24,25,26,27].

This pathogenetic model of depression has evolved since its initial formulation, including now adaptive changes in receptors and the participation of second messengers/signal transduction pathways (cAMP, phosphokinase A, cAMP-response element-binding protein, neurotrophin-mediated pathway, etc.), elements which are explored in connection with the original hypothesis [28,29]. Although heavily criticized from diverse perspectives, e.g., not explaining the latency of response to antidepressants, not answering the bothersome question of why there is still a high rate of nonresponse even after multiple trials of antidepressants, etc. [29,30], the monoamine hypothesis is still productive, and new agents are researched starting from its core concept [31]. Moreover, it is important to observe that new pathogenetic models for depression have been formulated in the last decades (e.g., based on glutamatergic neurotransmission and neuroplasticity), and new antidepressants, with a faster onset of their therapeutic action, have already been marketed, i.e., brexanolone and esketamine [29,32,33]. As observed by several authors, “in the era of neural networks and systems-level neuroscience, ‘single’ neurotransmitter theories of depression look increasingly implausible” [34].

Translational research is expected to play an important role in finding new pathophysiological models for MDD and, therefore, new therapeutic agents. In this context, preclinical research is essential for connecting theoretical knowledge with clinical data by exploring various paradigms, such as genetic models, acute/chronic stress models, brain neurotransmitters/specific brain injury models, and models induced by pharmacological interventions [35,36,37,38]. Although none of these models completely reproduce human MDD (i.e., face validity, construct validity, and predictive validity are still far from being perfect), they are considered relevant because of their capacity to mimic certain features of this pathological condition and thus allowing for the systematic investigation of etiology, pathogenesis, and treatment of depression [36,39]. For example, the antidepressant effects of kynurenic acid, an L-tryptophan metabolite with neuromodulatory activities, have been explored in preclinical studies [40]. In a modified forced swimming test in mice, kynurenic acid successfully reversed immobility, climbing, and swimming times [40]. Moreover, this study showed that the antidepressant effects of kynurenic acid are modeled by this pathway’s interactions with GABA-ergic (through GABA-A receptors), dopaminergic (i.e., D2, D3, and D4 receptors), and serotonergic (i.e., 5-HT2 receptors) neurotransmission [40]. The stress vulnerability model has been proven useful in separating depressive-like and anxious-like phenotypes during benzodiazepine administration in mice; this model has further supported the utility of brain-derived neurotrophic factor (BDNF) in developing neural and behavioral plasticity and the effects of antidepressants [39,41]. The chronic unpredictable mild stress model is another robust tool for the pharmacological research of MDD, and various antidepressants have been explored using this paradigm; for example, antidepressant-induced reduction in hippocampal cytogenesis has been correlated with symptoms amelioration in a study investigating the effects of escitalopram in rats with this chronic stress model [39,42].

Triple reuptake inhibitors (TRI) are a class of antidepressants that target the serotonin, norepinephrine, and dopamine neurotransmission simultaneously (Figure 1), thus representing a step forward in treating MDD, at least at the conceptual level [11,43]. It is expected that by administering these agents, the need to augment antidepressants that enhance the activity of only one or two monoaminergic systems (e.g., the selective serotonin reuptake inhibitors, or serotonin and norepinephrine reuptake inhibitors) would be less frequently met. In turn, this would present multiple possible advantages, such as the elimination of potentially negative pharmacokinetic interactions; decreasing the risks for adverse events or toxic reactions due to polypharmacy; reducing the latency of therapeutic effect by eliminating the need for a monotherapy trial followed by the addition of/switching to another agent; and mitigating the adverse effects due to serotonin activation by adding dopamine-activatory features, etc. (Figure 2).

Many agents within the TRI class have been described in the last two decades, some of them reaching the clinical stage of research while others being discontinued in the preclinical phase. An exhaustive presentation of these agents is beyond the aim of this review, and it has been carried out very well elsewhere [44,45]. It is also to be noted that TRIs have been explored in clinical trials for different indications, starting from MDD to attention-deficit hyperactivity disorder (ADHD), and from pain disorder to cocaine use disorder [44,45].

To illustrate the complexity of this pharmacological category, several agents investigated for their antidepressant effect will be presented further, emphasizing their pharmacodynamic profile and results from clinical/preclinical studies. For example, DOV 216,303 is a TRI with IC_50_ values of 14 nM, 20 nM, and 78 nM for serotonin, norepinephrine, and dopamine uptake, which was explored in healthy volunteers and was found safe and well tolerated; in a phase 2 study, this agent was compared with citalopram in patients with MDD and both antidepressants decreased the Hamilton Depression Rating Scale (HAMD) scores compared with the baseline; no placebo arm existed in this phase 2 trial [46].

DOV 21,947 (amitifadine, EB-1010) is the (+) enantiomer of DOV-216,303 and it is a serotonin-preferring TRI with a potency to block the reuptake of serotonin, norepinephrine, and dopamine of 12 nM, 23 nM, and 96 nM, respectively; in a phase 2 study, the change in Montgomery–Asberg Depression Rating Scale (MADRS) scores was significant vs. placebo at week 6, and the tolerability was good; and in a phase 2b/3a study, the efficacy was not confirmed in patients with MDD at 6 weeks [47,48].

Ro 8-4650 (Diclofensine) is a TRI with IC_50_ for serotonin, norepinephrine, and dopamine of 4.8 μM, 2.5 μM, and 4.5 μM, respectively; it was found to be well tolerated in studies with healthy volunteers and MDD patients, with an efficacy comparable to that of maprotiline [49,50].

SKF83959 is a competitive inhibitor of serotonin transporter and a non-competitive blocker of norepinephrine and dopamine transporters, with K_i_ of 1.43 ± 0.45 μmol/L, 0.6 ± 0.07 μmol/L, and 9.01 ± 0.8 μmol/L, respectively. In preclinical studies (i.e., the chronic social defeat stress model of depression), this molecule has proven antidepressant effects and also mitigated the decrease in hippocampal brain-derived neurotrophic factor (BDNF) signaling pathway, dendritic spine density, and neurogenesis induced by stress [51,52].

BMS-820836 is another TRI with IC_50_ values of 0.2 nM, 26.7 nM, and 6.19 nM for serotonin, norepinephrine, and dopamine transporters, respectively. In two randomized, phase 2b studies, this investigational product was well tolerated in patients with TRD during six weeks, but failed to show superiority versus continuation of an existing antidepressant (i.e., duloxetine or citalopram) [53,54].

Regarding the currently marketed antidepressants, which target all three monoamine transporters in different degrees, it is considered useful to compare at least some of them with TRIs from the perspective of their affinity for these transporters. For example, the inhibition of serotonin, norepinephrine, and dopamine transporters in the case of amitriptyline provides IC_50_ values of 67 nM, 63 nM, and 7500 nM, respectively; for duloxetine, the values are 13 nM, 42 nM, and 439 nM; for venlafaxine the values are 145 nM, 2483 nM, and 7647 nM; and for milnacipran the values are 151, 200, and >100,000, respectively [55]. In the case of desvenlafaxine, the IC_50_ was 47.3 ± 19.4 and 531.3 ± 113 nM for serotonin and norepinephrine transporters, respectively [56]. These distinct affinities for monoamine transporters are important for the clinical effects of these agents and explain their use not only in the treatment of MDD or TRD but also in fibromyalgia, diabetic peripheral neuropathy, migraine, anxiety disorders, etc. [57]. Moreover, these pharmacodynamic profiles have an impact on the tolerability of the explored drugs; for example, serotonergic activity enhancement may cause sexual dysfunctions, nausea, and sleep dysfunctions, while the activation of noradrenaline pathways can cause urinary hesitancy and insomnia, but are less likely to induce sexual dysfunction [7,58,59,60]. The activation of dopamine neurotransmission has been explored in relation to an addictive potential (which was found not significant for antidepressants administered in usual doses) [59].

Toludesvenlafaxine (LY03005, LPM570065, ansofaxine, and anshufaxine) is a TRI that is able to block the reuptake of serotonin, dopamine, and norepinephrine in the CNS [61,62]. The formula of this drug is 4-methyl benzoate desvenlafaxine hydrochloride, and it could be converted rapidly into desvenlafaxine in vivo, but both substances may coexist in the brain due to their liposolubility [62]. The product was authorized for marketing in China as toludesvenlafaxine hydrochloride by the National Medicinal Products Administration (NMPA) in November 2022, and it is the only available triple monoamine reuptake inhibitor (first-in-class product) [63]. As stated in a press release, “the application for marketing authorization is based on clinical data generated from six clinical studies in China” [64]. Moreover, a New Drug Application (NDA) file was reviewed and accepted by the Food and Drug Administration (FDA) in 2020 for LY03005, and a phase 1 study with this pharmacological agent has been completed in Japan [64,65]. Ansofaxine has been explored in clinical trials in the United States, Japan, the European Union, and China [66].

The importance of TRI investigation is derived from theoretical and pragmatic reasons. As previously mentioned, the monoamine hypothesis is under extensive scrutiny for its clinical limitations, and new theoretical frameworks for MDD have been formulated, based on other neurotransmitters’ dysfunction, gut microbiome alterations, inflammatory factors, immune or hypothalamic–pituitary–adrenal axis dysregulation, and neurogenesis or cholesterol biosynthesis pathway impairments [29,67,68,69,70,71]. Still, the monoaminergic system is considered a central element in the pathogenesis of MDD, although multiple interactions with other CNS pathways are also important for the understanding of this disorder [72,73]. At a practical level, despite quite a large armamentarium of pharmacological agents for MDD, there is an important percentage of patients who did not reach remission or have a high relapse rate [74,75]. Therefore, the validation through clinical research of new treatments for MDD is paramount for the quality of life, clinical evolution, and prognosis of these patients.

The main objective of this narrative review was to search for data regarding the pharmacological and clinical properties of a new TRI, toludesvenlafaxine. Efficacy, tolerability, and safety, as well as the pharmacological profile of this TRI, were considered important elements to integrate into the case management of patients with MDD or TRD.

A narrative review dedicated to finding evidence for the efficacy and tolerability of toludesvenlafaxine has been conducted by searching five electronic databases (PubMed, Cochrane, Clarivate/Web of Science, Google Scholar, and EMBASE) using the paradigm “toludesvenlafaxine” OR “ansofaxine” OR “LY03005” OR “LPM570065” AND “efficacy” OR “tolerability” OR “safety” OR “pharmacology” OR “clinical studies” OR “preclinical studies”. Moreover, references within the main reviewed papers were searched for supplementary information when it was considered consistent with the objective of this review.

Due to the novelty of the researched pharmacological agent, grey literature was included according to the Luxembourg definition [76]. More specifically, three sources of information were targeted: (a) main repositories of clinical trials run by the United States National Library of Medicine and the National Institutes of Health—www.clinicaltrials.gov (accessed on 26 December 2022), World Health Organization (International Clinical Trials Registry Platform)—www.who.int/clinical-trials-registry-platform (accessed on 26 December 2022), the European Union (EU Clinical Trial Register)—www.clinicaltrialsregister.eu (accessed on 26 December 202), and the Chinese Clinical Trial Registry (ChiCTR)—http://www.chictr.org.cn/index.aspx (accessed on 26 December 2022); and (b) industry and commercial press releases containing references to ansofaxine/toludesvenlafaxine; (c) annual reports, news articles, presentations, and other non-peer-reviewed materials. For the grey literature, all sources including references to “ansofaxine”, “LY03005”, “LPM570065”, or “toludesvenlafaxine” were explored.

No inferior time limit was established for the retrieved papers included in the review, while the superior limit was December 2022. Moreover, no limitations regarding the language of the reports, the study environment (inpatient, outpatient, or mixed milieu for clinical trials), the population characteristics, or the type of research (preclinical or clinical studies) were established. As exclusion criteria, (1) reports that could not be attributed with certainty to the individual author(s), research institutes, commercial entities, and governmental or non-governmental institution(s), and (2) sources not specifying clinical, preclinical, or pharmacological data about toludesvenlafaxine were excluded.

## 2. Preclinical and Clinical Data on the Efficacy, Tolerability, Safety, and Pharmacological Profile of Toludesvenlafaxine

Based on the analysis of the retrieved data, 5 preclinical studies and 12 clinical studies were identified in the literature. However, while all the preclinical studies had published results (Table 1), the clinical trials presented a more nuanced status—only two clinical trials had published results, five trials had results disclosed by the manufacturer, but not published in peer-reviewed journals, while another five trials had undisclosed results (Table 2).

### 2.1. Preclinical Studies

A preclinical study explored the acute (single dose administration) and long-term effects of LPM570065 in Sprague–Dawley rats, concluding the following: (a) the maximum tolerated dose in the acute-administration study was 500 mg/kg, and the lethal dose was 1000 mg/kg; (b) the 13-week study led to no significant adverse events in doses higher than 300 mg/kg for rats, with no mutagenic or clastogenic effects; (c) the maximum tolerated dose in clinical conditions was deduced to be 300 mg/day; and (d) the sexual functioning in patients who might receive this agent should be monitored because changes in the prolactin and testosterone levels were detected in this preclinical study [77].

Another preclinical study explored the effects of LPM570065 on fertility and early embryonic development in Sprague–Dawley rats, concluding that (a) no observable adverse effect level was established at 100 mg (female rats) and 300 mg/kg (male rats); (b) the same parameter determined this time for fertility and early embryonic development was established at 300 mg/kg (female rats) and 100 mg/kg (male rats) [78].

In another preclinical study, acute administration of LPM570065 or desvenlafaxine was initiated in rats using an oral or intravenous solution [79]. High-performance liquid chromatography analysis showed that ansofaxine can rapidly penetrate the striatum and is converted into desvenlafaxine while presenting larger total exposure vs. desvenlafaxine [79]. Long-term administration of ansofaxine (up to 14 days) via the oral route increased all three monoamine levels more than desvenlafaxine, especially dopamine levels (detected by microdialysis) [79]. During the forced swim test, acute and chronic administration of ansofaxine decreased the immobility time more than desvenlafaxine, suggesting a higher efficacy and/or a more rapid onset of antidepressant effect than desvenlafaxine [79].

In a “two-hit” stress mouse model (early life maternal separation and social defeat stress), three behavioral models were applied—sucrose preference test, tail suspension test, and forced swimming test—in adult mice receiving LPM570065 [80]. According to this animal study, ansofaxine significantly reversed depressive-like behaviors in all three paradigms used and increased the density of dendritic spines in the CA1 hippocampal neurons [80]. LPM570065 also reduced the hypermethylation of the oxytocin receptor gene (*Oxtr*) in the hippocampus of mice experiencing the “two hits” stress [80]. This last observation suggests that LPM570065 may reduce depression vulnerability via epigenetic mechanisms involving the *Oxtr* expression [80].

Toludesvenlafaxine has a high binding affinity for serotonin, norepinephrine, and dopamine transporters and significantly inhibited the reuptake of all three monoamines- IC_50_ = 31.4 ± 0.4 nM for serotonin, IC_50_ = 586.7 ± 83.6 nM for norepinephrine, and IC_50_ = 733.2 ± 10.3 nM for dopamine in vitro (serotonin:norepinephrine:dopamine = 23.3:1.2:1) [46]. The highest inhibition for serotonin transporters was reported in in vitro assays [46]. The antidepressant effects were observed in rodent models at 8–16 mg/kg [46]. The absorption was good after oral administration, and it was converted to O-desvenlafaxine due to the action of esterases in vivo, both reaching the hypothalamus in high concentration [81]. The plasma exposure was proportional to the dose after oral administration [81]. While desvenlafaxine does not increase the striatal level of dopamine, toludesvenlafaxine has this effect, which indicates supplementary benefits vs. the older drug [81]. The preclinical data do not support the existence of CNS excessive activation (manifested as irritation or hyperreactivity)/depression (reflected in sedation or muscle relaxation) and no significant abuse potential [77,81].

**Table 1 pharmaceuticals-16-00411-t001:** Summary of the preclinical studies focused on toludesvenlafaxine.

Design	Results	Observations	Reference
Single and 13-week repeated-dose oral toxicity assessment and mutagenicity assays. Acute dose: 500 mg/kg, 1000 mg/kg, and 2000 mg/kg LPM570065 in SD rats. A 13-week toxicity study: 30 mg/kg, 100 mg/kg, or 300 mg/kg LPM570065 for 13 consecutive weeks + 4-week recovery period. N = 80 rats (40 males and 40 females).	In a single-dose acute study: 2 out of 20 rats died in the 1000 mg/kg group vs. seven out of 20 in the 2000 mg/kg group vs. none in the 500 mg/kg group. In the 13-week toxicity study: transitory salivation and minor body weight decrease was reported in the 300 mg/kg group in males. Serum PRL levels ↓ by 43% and 78% in male rats in 100 mg/kg and 300 mg/kg groups, respectively. Serum TST ↑ by 37% in the 30 mg/kg and 100 mg/kg males.	MTD = 500 mg/kg and lethal dose = 1000 mg/kg in the acute administration. In the long-term administration, no observed AE level was ≥300 mg/kg for rats; no mutagenic or clastogenic effects. MTD = 3000 mg/patient/day in clinical conditions. The effects of LPM570065 on sexual function are to be monitored.	Li C, Jiang W, Gao Y, et al. [77]
Acute phase: 30 mg/kg, 100 mg/kg, and 300 mg/kg LPM570065 vs. control. Female rats received 2 weeks of the investigational product + mating up to the 7th gestation day. Male rats received 4 weeks of investigational product + mating with treated female rats. Following this stage, all males were treated up to the ninth week and a new mating period was initiated with non-treated female rats. Mortality, toxicity symptoms, body weight, amount of food consumed, sexual cycle, mating behavior, pregnancy, sperm production, gross necropsy, and weight of organs. N = 264 rats were distributed in 4 groups (44 females and 22 males in each group).	Excessive salivation post-treatment in all females and males on 100 mg/kg and 300 mg/kg LPM570065 groups. BW gain ↓ in gravid rats with 300 mg/kg investigational product during gestation days 0–6. Decreased fertility rates were associated with a 300 mg/kg dose of investigational product in male rates. Sperm concentration and count were higher in all three groups treated with LPM570065 vs. controls. Duration of mating ↓ significantly to 37.5% after 9 weeks of treatment with 300 mg/kg.	The no observable AE level was established at 100 mg/kg (female rats) and 300 mg/kg (male rats). The no observable AE level for fertility and early embryonic development was established at 300 mg/kg (female rats) and 100 mg/kg (male rats).	Guo W, Gao Y, Jiang W, et al. [78]
Exploring extracellular 5HT, NE, and DA levels in the rat striatum after acute and chronic administration of LPM570065 vs. DSVLFX. The methods used were HPLC and microdialysis. N = 72 rats divided into 9 equal groups.	HPLC results showed that LPM570065 rapidly penetrates the striatum and converts into DSVLFX while presenting larger total exposure vs. DSVLFX. Long-term administration of LPM570065 (up to 14 days) via the oral route increased all three monoamine levels more than DSVLFX, and especially dopamine levels (detected by microdialysis). During the forced swim test, acute and chronic administration of LPM570065 ↓ the immobility time more than DSVLFX.	LPM570065 may possess an ↑ efficacy and/or a more rapid onset of antidepressant effect than DSVLFX. LPM570065 counterbalances the negative effects of DSVLFX on 5HT neurotransmission related to the 5HT1A autoreceptors.	Zhang R, Li X, Shi Y, et al. [79]
Adult male and female C57BL/6J mice, 5 groups, each group had 24 animals: control vs. single-stress vs. double-stress vs. LPM570065 vs. fluoxetine groups. Sucrose preference test, forced swimming test, and tail suspension test.	LPM570065 reduced susceptibility to depression-like behaviors in adult mice + maternal separation. LPM570065 protected against the reduced number of dendritic spines in the hippocampal CA1 of mice subjected to stress. LPM570065 regulated the expression of DNMTs in the mouse hippocampus.	LPM570065 may reduce depression vulnerability via epigenetic mechanisms involving the *Oxtr* expression.	Meng P, Li C, Duan S, et al. [80]
Male and female Wistar and Sprague–Dawley rats (total of 12/sex/group and 5/sex/group, respectively); affinity for monoamine transporters was determined by radioligand membrane binding assay; chronic unpredictable mild stress procedure; rat olfactory bulbectomized model, open field test, sucrose consumption test, serum corticosterone, and testosterone levels. Toludesvenlafaxine 10 μM.	The highest inhibition for serotonin transporters was reported in in vitro assays. The absorption was good after oral administration, and it was converted to O-desvenlafaxine due to the action of esterases in vivo, both reaching the hypothalamus in high concentration.	While desvenlafaxine does not increase the striatal level of dopamine, toludesvenlafaxine has this effect, which indicates supplementary benefits vs. the older drug.	Zhu H, Wang W, Sha C, et al. [81]

5HT = serotonin; AE = adverse effect; BW = body weight; DA = dopamine; DNMT = DNA methyltransferases; DSVLFX = desvenlafaxine; HPLC = high-performance liquid chromatography; MTD = maximum tolerated dose; NE = norepinephrine; *Oxtr* = oxytocin receptor; PRL = prolactine; SD = Sprague–Dawley; and TST = testosterone.

### 2.2. Clinical Trials

Out of the 12 references found for clinical trials, from phase 1 to 3, only a phase 2 and a phase 3 study had published results [62,82]. A total of 5 phase 1 trials conducted in different sites in the U.S. and China had results presented in manufacturer press releases [83,84,85,86]. Another 5 phase 1 trials had undisclosed results [64,87,88,89,90].

A total of 3 phase 1 studies were conducted in China: a single dose ascending trial (N = 72 healthy volunteers) explored the pharmacokinetics and tolerability of 20–200 mg LY03005, another study (N = 12 subjects) investigated the effects of food on the pharmacokinetics of 120 mg investigational product, and yet another (N = 48 participants) received multiple ascending doses of LY03005 [85]. No significant effect of the food on the bioavailability was observed, and the concentrations of the main active metabolite of LY03005 were dose-proportional for the 20–200 mg range [85]. The multiple-dose ascending dose study concluded that the steady state of the main active metabolite could be reached on the third day after daily dosing, and the concentrations of this metabolite were dose-proportional at the steady state for the dose of LY03005 ranging from 40–160 mg/day [85]. A good safety and tolerability profile of LY03005 was supported by all these trials (two main trials and a substudy) [85]. Besides these studies, another 2 phase 1 trials took place in the U.S. and enrolled a total of 120 healthy volunteers [48,49,51]. In the randomized, double-blind, single-ascending dose study, 72 subjects received 20 mg, 40 mg, 80 mg, 120 mg, 160 mg, or 200 mg LY03005 or placebo, and the results supported a good safety profile and linear dose proportionality on the plasma exposure after a unique dose of investigational product [84,86]. A substudy explored the effect of food on the pharmacokinetics of LY03005 in 10 subjects, but no impact on the bioavailability was detected [84,86]. In the multiple-ascending-dose, 48 healthy volunteers received daily 1 of the 4 regimens—40 mg, 80 mg, 120 mg, or 160—or placebo for 8 consecutive days [83,86]. The overall safety profile was good and linear dose proportionality on the plasma exposure was confirmed after repeated dose administration; the steady state of plasma exposure was reached after the third or fourth oral intake of the investigational product [83,86].

Orally administered ansofaxine extended-release (ER) was explored in a phase 2, multicenter, randomized, double-blind, placebo-controlled, dose-finding trial, which enrolled 260 patients with MDD (18–65 years old) [62]. Fixed doses of ansofaxine, i.e., 40 mg/day, 80 mg/day, 120 mg/day, or 160 mg/day or placebo, were administered for 6 weeks, and the primary outcome measure was the change in total HAMD-17 items from baseline to week 6 [62]. At the endpoint, significant changes were reported in all the groups that received the active drug vs. placebo, and the tolerability of all doses of ansofaxine was good [62]. The incidence of treatment-related adverse events was 52% (vs. 38.8% in the placebo group), reported by 141 patients and totalizing 303 cases [62].

**Table 2 pharmaceuticals-16-00411-t002:** Registered clinical trials exploring the efficacy and/or tolerability of toludesvenlafaxine (ansofaxine).

Methodology	Primary Outcome(s) and Measures	Secondary Outcome(s) and Measures	Sponsor of the Clinical Trial	The Country Where the Clinical Trial Took Place	Status of the Trial	Results and Observations	Registration of Clinical Trial and/or Reference(s)
LY03005 (40 mg, 80 mg, 120 mg, and 160 mg) vs. placebo, DBRCT, phase 2, dose-finding study, N = 260 MDD patients (18–65 years old), 2 weeks wash out + 6 weeks treatment	HAMD-17 scores at week 8	MADRS and CGI-I at week 8	Luye Pharma Group Ltd. (China)	China	Completed	HAMD-17 scores were significantly changed by the intervention vs. placebo at week 6 in all active treatment groups vs. placebo (*p* < 0.05). All doses were generally well tolerated, but the % of AEs was superior to the placebo group in each active treatment group.	NCT03785652 [62,91]
LY03005 (80 mg or 160 mg) vs. placebo, DBRCT, phase 3, N = 558 MDD patients (18–65 years old), 1-week screening + 8-week double-blind treatment	MADRS scores at 8 weeks	HAMD-17 at week 8	Luye Pharma Group Ltd. (China)	China	Completed	HAMD-17 total score and “anxiety/somatization”, “cognitive impairment”, and “blocking” factors, CGI, HAM-A, SDS, and MADRS “anhedonia factor” scores were significantly improved vs. placebo at week 8. Most of the adverse events were mild and moderate, and no SAE was reported. Nausea, vomiting, headache, and drowsiness were the most frequently reported (over 5%) AEs in the active treatment groups.	NCT04853407 [82,92]
LY03005 (20 mg, 40 mg, 80 mg, 120 mg, 160 mg, 200 mg, and 120 mg + fed) vs. DSVLFX (50 mg) vs. placebo, phase 1, RDBCT, N = 72 healthy participants in the SAD study + 12 subjects in food effect study (18–45 years old)	Number of participants with AEs during 11 days	PK parameters-C_max_ up to 4 days	Luye Pharma Group Ltd. (China)	United States	Completed	Unpublished results. No obvious effect of food on the bioavailability of LY03005. Good safety profile and linear dose proportionality on the plasma exposure after single oral dose administration.	NCT02055300 [84,86]
LY03005 (40 mg, 80 mg, 120 mg, or 160 mg) vs. placebo, phase 1, RDBCT, MAD, N = 48 healthy subjects (18–45 years old), 8 consecutive days	Number of participants with AEs during 3 to 4 months	The PK of MAD	Luye Pharma Group Ltd. (China)	United States	Completed	Unpublished results. Good safety profile of LY03005 treatment and linear dose proportionality on the plasma exposure after multiple oral administrations. The steady state of plasma exposure was reached after 3rd or 4th oral intake of the investigational product.	NCT02271412 [83,86]
Phase 1 trials, healthy volunteers (N = 132), single or multiple doses of oral LY03005 administration; N_1_ = 72 subjects in SAD study, N_2_ = 12 subjects in the food-effect study, and N_3_ = 48 subjects in the MAD study.	Safety and PK profiles for LY03005		Luye Pharma Group Ltd. (China)	China	Completed	Unpublished results. SAD study: concentrations of the main active metabolite were dose proportional for the dose range of 20–200 mg LY03005. Food-effect study: food did not affect the bioavailability in healthy subjects. MAD study: the steady state of the main active metabolite could be achieved on the 3rd day following multiple dosing; concentrations of the main active metabolite were dose proportional at the steady state for the dose range 40–160 mg/day LY03005. All studies: good tolerability and safety.	CTR20130364, CTR20140333, and CTR20140418 [85]
LY03005 (80 mg) vs. DSVLFX (50 mg), phase 1, pilot study, open-label study, single dose, N = 20 healthy subjects (18–50 years old)	Bioavailability of oral tablets under fasting conditions: AUC-PK samples were drawn at t0 (i.e., 30 min prior to dosing), 1 h, 2 h, 3 h, 4 h, 6 h, 8 h, 10 h, 12 h, 23 h, 32 h, 48 h, and 72 h after dosing		Luye Pharma Group Ltd. (China)	United States	Completed	Undisclosed	NCT02988024 [87]
LY03005 (80 mg) fasted vs. fed crossover open-label randomized trial, single dose, phase 1, N = 34 participants (18–50 years old)	AUC and Cmax:PK parameters (predose and after dose) of parent and active metabolite		Luye Pharma Group Ltd. (China)	United States	Completed	Undisclosed	NCT03822065 [88]
LY03005 (80 mg) vs. DSVLFX (50 mg) 2-sequence, 2-period crossover open-label randomized trial, phase 1, single dose, N = 56 healthy participants (18–50 years)	AUC 15 days, parent drug and its active metabolite		Luye Pharma Group Ltd. (China)	United States	Completed	Undisclosed	NCT03733574 [89]
LY03005 (80 mg) vs. DSVLFX (50 mg), phase 1, randomized, open-label, cross-over, 2-period study, single dose, N = 20 healthy participants (18–50 years old)	AUC assessment up to 72 h after dosing in both trial periods. C_max_ assessment up to 72 h after dosing in both trial periods.	AEs assessment up to 35 days	Luye Pharma Group Ltd. (China)	United States	Completed	Undisclosed	NCT03357796 [90]
Phase 1 trial, healthy volunteers	Safety and tolerability of LY03005		Luye Pharma Group Ltd. (China)	Japan	Completed	Undisclosed	[64]

AEs = adverse events; AUC = area under the curve; CGI-I = Clinical Global Impression Scale—Improvement; DSVLFX = desvenlafaxine; HAMA = Hamilton Anxiety Scale; HAMD-17 = Hamilton Depression Rating Scale—17-item version; MAD = multiple ascending dose; MADRS = Montgomery–Asberg Depression Rating Scale; MDD = major depressive disorder; PK = pharmacokinetics; RDBCT = randomized double-blind controlled trial; SAE = severe adverse events; SAD = single ascending dose; and SDS = Sheehan Disability Scale.

In a phase 3 trial conducted by the manufacturer in China, which enrolled 558 adults with MDD (according to the DSM-5 criteria), ansofaxine doses of 80 mg and 160 mg were compared with placebo, and efficacy and tolerability parameters were monitored for 8 weeks [82]. This double-blind, randomized trial showed significant improvement in primary and secondary outcomes in patients who received the active drug vs. placebo at the end point [82]. Montgomery–Asberg Depression Scale (MADRS) scores decreased significantly at week 8 vs. baseline values and vs. placebo in the 2 groups which received 80 mg and 160 mg of active drug, respectively [82]. HAMD-17 items total score and “anxiety/somatization”, “cognitive impairment”, and “blocking” factors, Clinical Global Impression (CGI), Hamilton Anxiety Scale (HAM-A), Sheehan Disability Scale (SDS), and MADRS “anhedonia factor” scores were also significantly improved vs. placebo at the end of week 8 [82]. Most of the adverse events were mild and moderate, and no serious adverse event was reported [82]. Nausea, vomiting, headache, and drowsiness were the most frequently reported (over 5%) adverse events in the active treatment groups [82].

## 3. Discussion

The findings of this review indicate there are practical and theoretical reasons for further exploration of the clinical properties of toludesvenlafaxine in larger and longer trials, focused both on efficacy and tolerability. The available data seem promising; therefore, this further exploration is expected to increase our knowledge of the TRI utility in MDD patients and would be expected to improve their chances of recovery. However, this process is not without challenges since the monoaminergic hypothesis is undergoing extensive criticism, and new pathogenetic mechanisms have already produced clinically available antidepressants, as mentioned in the introductory chapter. Still, the field of antidepressant research is a very intensively explored one, and there are multiple reasons to expect this interest will not decrease any time soon. The social and economic burden of treatment-resistant MDD makes the subject of finding new antidepressants very urgent [93,94,95,96].

Based on the reviewed data, toludesvenlafaxine possesses a high affinity for all three monoaminergic transporters, and preclinical data showed increased levels of serotonin, dopamine, and norepinephrine in rat striatum after acute or chronic administration of this drug [62,79]. A pharmacodynamic comparison between toludesvenlafaxine and other antidepressants that enhance serotonergic and norepinephrinergic ± dopaminergic neurotransmission is presented in Table 3.

Changes in prolactin and testosterone levels were reported in preclinical studies [77]. No significant adverse effects were observed, including mutagenic and clastogenic events up to 13 weeks and above 300 mg/day in rats [77]. Fertility and early embryonic development were not affected by the investigational product at doses up to 300 mg/kg for female rats and 100 mg/kg for male rats [78]. Acute injection of toludesvenlafaxine led to larger exposure vs. desvenlafaxine, and long-term administration of this agent (up to two weeks) increased all three monoamines more than desvenlafaxine, but especially dopamine [79]. A higher efficacy and/or more rapid onset of antidepressant effect than desvenlafaxine was also deduced based on the preclinical models of depression [79]. Toludesvenlafaxine significantly reversed depressive-like behaviors in all three paradigms used and increased the density of dendritic spines in the CA1 hippocampal neurons and reduced the hypermethylation of the oxytocin receptor gene (*Oxtr*) in the hippocampus of mice experiencing the “two hits” stress; therefore, it may reduce depression vulnerability via epigenetic mechanisms involving the *Oxtr* expression [80]. The plasma exposure was proportional to the dose after oral administration [81]. While desvenlafaxine does not increase the striatal level of dopamine, toludesvenlafaxine has this effect, which indicates supplementary benefits vs. the older drug [81]. The preclinical data do not support the existence of CNS excessive activation (manifested as irritation or hyperreactivity)/depression (reflected in sedation or muscle relaxation) and no significant abuse potential [77,81].

All phase 1 studies conducted in China and the U.S. supported a good tolerability and safety profile without a significant effect of the food on the bioavailability, and the concentrations of the main active metabolite of LY03005 were dose proportional for the 20–200 mg range [83,84,85,86]. The steady state of the main active metabolite could be reached on the third day after daily dosing, and the concentrations of this metabolite were dose-proportional at the steady state for the dose of LY03005, ranging from 40–160 mg/day [85].

Orally administered toludesvenlafaxine extended-release (ER) was efficient in a phase 2 study and significantly decreased the HAMD-17 scores after 6 weeks in patients with MDD, while its tolerability was good [62]. In a phase 3 trial, toludesvenlafaxine reduced MADRS scores significantly at week 8 vs. baseline values and vs. placebo after 8 weeks in the 2 groups which received 80 mg and 160 mg of active drug, respectively [82]. HAMD-17 items total score and “anxiety/somatization”, “cognitive impairment”, and “blocking” factors, CGI, HAM-A, SDS, and MADRS “anhedonia factor” scores were also significantly improved vs. placebo at the end of week 8 [82]. Most of the adverse events were mild and moderate, and no serious adverse event was reported [82].

**Table 3 pharmaceuticals-16-00411-t003:** Pharmacodynamic characteristics of TRIs and other antidepressants [54,55,56,81].

Pharmacological Agent	SERT	NET	DAT	Observations
Toludesvenlafaxine	IC_50_ = 31.4 ± 0.4 nM	IC_50_ = 586.7 ± 83.6 nM	IC_50_ = 733.2 ± 10.3 nM	Prodrug of desvenlafaxine, TRI
Desvenlafaxine	IC_50_ = 47.3 ± 19.4 nM	IC_50_ = 531.3 ± 113 nM	-	Major metabolite of venlafaxine, SNRI
Venlafaxine	IC_50_ = 145 nM	IC_50_ = 2483 nM	IC_50_ = 7647 nM	SNRI
Duloxetine	IC_50_ = 13 nM	IC_50_ = 42 nM	IC_50_ = 439 nM	SNRI
Milnacipran	IC_50_ = 151 nM	IC_50_ = 200 nM	IC_50_ > 100,000	SNRI
DOV 216,303	IC_50_ = 14 nM	IC_50_ = 20 nM	IC_50_ = 78 nM	TRI
DOV 21,947	IC_50_ = 12 nM	IC_50_ = 23 nM	IC_50_ = 96 nM	(+)-DOV-216,303, TRI
Ro 8-4650	IC_50_ = 4.8 μM	IC_50_ = 2.5 μM	IC_50_ = 4.5 μM	TRI
SKF83959	K_i_ = 1.43 ± 0.45 μmol/L	K_i_ = 0.6 ± 0.07 μmol/L	K_i_ = 9.01 ± 0.8 μmol/L	TRI
BMS-820836	IC_50_ = 0.2	IC_50_ = 26.7 nM	IC_50_ = 6.19 nM	TRI
Amitriptyline	IC_50_ = 67 nM	IC_50_ = 63 nM	IC_50_ = 7500 nM	Tricyclic antidepressant

DAT = dopamine transporter; NET = norepinephrine transporter; SERT = serotonin transporter; SNRI = serotonin and norepinephrine reuptake inhibitor; and TRI = triple reuptake inhibitor.

Medication-specific aspects that have been correlated with treatment adherence were adverse effects, delayed onset of action, and subtherapeutic doses, but also complicated dosing schedules or titration strategies [97,98]. It is expected that TRI will associate a lower burden associated with pharmacological treatment with a lower risk of pharmacokinetic interactions, better tolerability, and higher treatment adherence because fewer drugs will be administered for the enhancement of all three monoamine transmitters. Moreover, a higher impact on dopamine neurotransmission observed in the case of toludesvenlafaxine could be beneficial in patients with MDD and substance use disorders, hyposexual desire disorder, or serotonin-induced sexual dysfunctions [44,45,64].

The main advantage of this review is the novelty of the topic because no other synthesis of preclinical and clinical data on toludesvenlafaxine has been identified in the literature during the initial search. All retrieved data on efficacy, tolerability, safety, and clinical pharmacology were analyzed, regardless of the research stage.

As a limitation of this review, it should be mentioned that it was only narrative, which could be considered a minus. However, because a broad search paradigm was used and extensive sources (white and grey) were included, it is not expected that important reports on toludesvenlafaxine were missed. An important number of trials did not have published results, and yet another study had undisclosed results. However, the major limitation of this review is represented by the fact that only two short-term trials that enrolled MDD patients were found [62,63], indicating that further research is needed in order to confirm the clinical utility of toludesvenlafaxine.

Future perspectives are related to the need to find the most appropriate target population for this antidepressant, following the principles of personalized medicine. For this purpose, pharmacogenetic studies are needed and comparative trials with other antidepressants in MDD patients are expected to provide the proper information for delineating the pharmacodynamic unicity of toludesvenlafaxine. Moreover, it is worth exploration of additional mechanisms of action for toludesvenlafaxine through translational research, in order to verify if enhancing the three monoaminergic pathways is related to other factors, such as immune or inflammatory markers, neurotrophic factors concentration, etc. Moreover, exploration of toludesvenlafaxine’s potential use in other psychiatric disorders, besides MDD, which share monoaminergic dysfunctions, e.g., anxiety disorders, fibromyalgia, obsessive-compulsive spectrum disorders, etc. [99,100], is considered to be highly important.

## 4. Conclusions and Future Perspectives

Toludesvenlafaxine is a first-in-class TRI that became available recently on the market in China. It is important to note the theoretical importance of toludesvenlafaxine due to its mechanism of action, in a moment when the monoaminergic hypothesis of depression is considered obsolete by part of the scientific community [29,34,101,102,103]. There are many hopes related to each new class of antidepressants that has been launched in the last decade, i.e., glutamatergic modulators, such as esketamine for treatment-resistant depression as an add-on to ongoing agents, or brexanolone for post-partum depression [104,105,106,107]. Although many other TRI agents have been explored, until now the results have not been encouraging, highlighting the importance of toludesvenlafaxine. Available data about the efficacy and tolerability of this antidepressant are encouraging, but it is still too early to fully describe its clinical properties because of the lack of long-term studies. It is expected that toludesvenlafaxine will improve the functional outcome of patients diagnosed with MDD and treatment-resistant MDD if the efficacy and safety of this new antidepressant will be confirmed by further translational and clinical research. This expectation is related to the fact that the simultaneous activation of all three monoaminergic systems (i.e., serotonin, dopamine, and norepinephrine) will have a favorable impact on mood (e.g., anhedonia, irritability, and depressive affect), cognitive (e.g., reduced attention, impaired learning, and memory deficits) and somatic (e.g., fatigue, loss of energy, and insomnia) dimensions of the depression.

## Figures and Tables

**Figure 1 pharmaceuticals-16-00411-f001:**
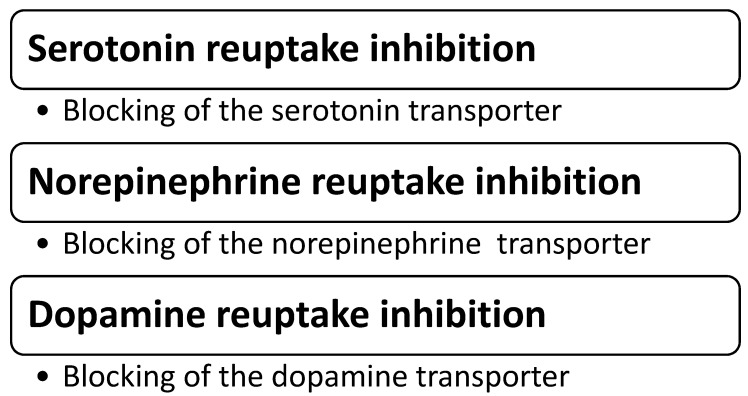
Mechanisms of action for TRI as antidepressants.

**Figure 2 pharmaceuticals-16-00411-f002:**
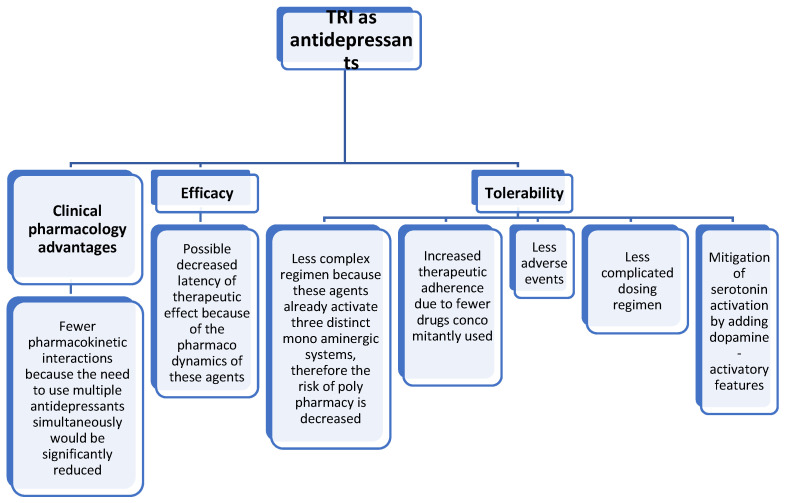
Potential benefits of TRI as antidepressants use in clinical practice.

## Data Availability

Not applicable.

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
