# Peer review of "Efficacy, Tolerability, and Safety of Toludesvenlafaxine for the Treatment of Major Depressive Disorder—A Narrative Review"

_pharmaceuticals, 2023, doi:10.3390/ph16030411_

Round 1
Reviewer 1 Report
1. Please edit the entire manuscript and use scientific words to make your idea better understandable. Rephrasing some of the sentences lost the meaning of your idea
2. Try to reduce some content from introduction part and emphasize more on your idea of Toludesvenlafaxine
3. Results are presented in a good way but you did not added discussion part instead you discussed your work under the heading of conclusion.
4. Discussion should be done separately and it need to be improved. Add some more previously published work to validate your hypothesis.
5. You added Table 3 in conclusion part. It seems awkward. Manuscript requires a thorough formatting according to journal guidelines.
Author Response
Dear Reviewer,
Thank you very much for the time and effort invested in reviewing this article. Kindly note my answers below. All modifications to the original manuscript are highlighted in yellow.
Q1. Please edit the entire manuscript and use scientific words to make your idea better understandable. Rephrasing some of the sentences lost the meaning of your idea.
A1. Rephrasing was done wherever it was observed as being necessary, and new fragments of text were added to support the core ideas.
Q2. Try to reduce some content from introduction part and emphasize more on your idea of Toludesvenlafaxine.
A2. There were some conflicting recommendations from Reviewers and Academic Editor regarding this aspect, therefore an extension was preferred, but with more focus on the TRI properties and debates regarding the monoaminergic hypothesis that supports the clinical utility of toludesveblafaxine.
Q3. Results are presented in a good way but you did not added discussion part instead you discussed your work under the heading of conclusion.
A3. A „Discussion” section was added.
Q4. Discussion should be done separately and it need to be improved. Add some more previously published work to validate your hypothesis.
A4. The „Discussion”section was formed by extracting from the „Conclusion”section the most relevant data, and by adding new information.
Q5. You added Table 3 in the conclusion part. It seems awkward. Manuscript requires a thorough formatting according to journal guidelines.
A5. Table 3 was removed in the „Discussion” part. References were formatted according to the journal’s guidelines.
Reviewer 2 Report
Dear Author.
The comments after reviewing the MS entitled “Toludesvenlafaxine for the treatment of the major depressive disorder” has been provided as under, point by point.
1. General comments
Apart from the minor mistakes in sentence structure, Abstract, keywords are correct. Introduction is sufficient and up to date. Adjustments in the figures and of tables are needed. Methodology is simple. The results of such clinical trials are organized and supported by relevant literature. Over all, the MS seems to be concise. References need to be according to the format of MDPI.
2. specific comments
· Page 2, figure 1. Corrections needed. The figure is either unnecessary or should be meaning full by using a systematic tree instead. Either the arguments should provide connectivity amongst them.
· page 4, methods. Correct the font sizes.
· page 5, table 1. References should be the last row and place only citation in order to reduce the size of table. Also check the format of tables 2. Make sure that all the tables are having same format.
· An improved graph representation must be provided that show statistical differences amongst polypharmacy in terms of activating multiple neurotransmitters pathways.
Author Response
Dear Reviewer,
Thank you very much for your thoughtful observations and recommendations. Kindly note my answers below. All modifications to the original manuscript are highlighted in yellow.
Q1. General comments
Apart from the minor mistakes in sentence structure, Abstract, keywords are correct. Introduction is sufficient and up to date. Adjustments in the figures and of tables are needed. Methodology is simple. The results of such clinical trials are organized and supported by relevant literature. Over all, the MS seems to be concise. References need to be according to the format of MDPI.
A1. References have been formatted according to the journal’s guidelines.
Q2. Specific comments
- Page 2, figure 1. Corrections needed. The figure is either unnecessary or should be meaning full by using a systematic tree instead. Either the arguments should provide connectivity amongst them.
Q2.1. A systematic tree was added instead of the previous listing-format figure.
- page 4, methods. Correct the font sizes.
Q2.2. The font size was corrected.
- page 5, table 1. References should be the last row and place only citation in order to reduce the size of table. Also check the format of tables 2. Make sure that all the tables are having same format.
Q2.3. References were moved in the last row. For consistency across tables 1 and 2, the source was preserved together with the citation. It was considered useful to preserve the NCTs in the second table, for readers who may want to see the full informations on the clinicaltrials.gov site. The format was changed, also for consistency across the three tables.
- An improved graph representation must be provided that show statistical differences amongst polypharmacy in terms of activating multiple neurotransmitters pathways.
Q2.4. Unfortunately, no data were retrieved with statistical significance between available antidepressants enhancing one vs. multiple neurotransmitters pathways. This is probably because other factors modulate the effects of antidepressants as well, e.g. BDNF, interleukins, cortisol concentrations etc.
Reviewer 3 Report
The review article “Toludesvenlafaxine for the treatment of the major depressive disorder”, provided a narrative for data regarding the pharmacological and clinical properties of a new TRI, toludesvenlafaxine.
Comments:
1. The authors should either mention ansofaxine, LY03005, LPM570065, and triple monoamine reuptake inhibitors in the abstract or remove them from the keywords section for consistency.
2. The information in Figure 1 could be better presented in the text as a description, rather than as a separate figure.
3. The conclusions section should focus on the review's conclusions. All those findings in the conclusions can be presented in the results section. Table 3 should be relocated to the results section for clarity.
4. The authors should aim to include the most recent references, ideally those published within the last 5 years, for a more up-to-date and accurate review.
Author Response
Dear Reviewer,
Thank you very much for your thoughtful observations and recommendations. Kindly note my answers below. All modifications to the original manuscript are highlighted in yellow.
Q1. The authors should either mention ansofaxine, LY03005, LPM570065, and triple monoamine reuptake inhibitors in the abstract or remove them from the keywords section for consistency.
A1. All the names of ansofaxine are now mentioned in the Abstract.
Q2. The information in Figure 1 could be better presented in the text as a description, rather than as a separate figure.
A2. There were multiple recommendations regarding this figure, so I choose to transform it into a logical tree. Details about the content of this figure are inserted in the text.
Q3. The conclusions section should focus on the review's conclusions. All those findings in the conclusions can be presented in the results section. Table 3 should be relocated to the results section for clarity.
A3. All the Reviewer’s suggestions led to the restructuring of the final part of the article.
Q4. The authors should aim to include the most recent references, ideally those published within the last 5 years, for a more up-to-date and accurate review.
A4. All available and relevant supporting papers were cited. The reference number increased from 59 to 79.
Reviewer 4 Report
The manuscript is very well written linguistically and presented with a solid review of the relevant literature and further research directions. It is an important research topic, and I believe it could lead to important clinical applications. I have a few suggestions.
The review should add a schematic representation of the mechanisms of action of triple reuptake inhibitors in the treatment of major depressive disorder.
Table 1 should be reorganized.
Author Response
Dear Reviewer,
Thank you very much for your thoughtful observations and recommendations. Kindly note my answers below. All modifications to the original manuscript are highlighted in yellow.
Q1. The review should add a schematic representation of the mechanisms of action of triple reuptake inhibitors in the treatment of major depressive disorder.
A1. Fig.1 is now dedicated to this subject.
Q2. Table 1 should be reorganized.
A2. The table was reorganized and re-formatted.
Round 2
Reviewer 1 Report
The authors have made all the requested changes
Author Response
Thank you again very much for your recommendations. I also wish to thank you for your feedback.